# Brain Protection after Anoxic Brain Injury: Is Lactate Supplementation Helpful?

**DOI:** 10.3390/cells10071714

**Published:** 2021-07-06

**Authors:** Filippo Annoni, Lorenzo Peluso, Elisa Gouvêa Bogossian, Jacques Creteur, Elisa R. Zanier, Fabio Silvio Taccone

**Affiliations:** 1Department of Intensive Care, Erasme Hospital, Free University of Brussels, Route de Lennik 808, 1070 Anderlecht, Belgium; lorenzo.peluso@ulb.be (L.P.); elisa.gouvea.bogossian@ulb.ac.be (E.G.B.); jacques.creteur@erasme.ulb.ac.be (J.C.); fabio.taccone@erasme.ulb.ac.be (F.S.T.); 2Laboratory of Acute Brain Injury and Therapeutic Strategies, Department of Neuroscience, Mario Negri Institute for Pharmacological Research IRCCS, Via Mario Negri 2, 20156 Milan, Italy; elisa.zanier@marionegri.it

**Keywords:** cardiac arrest, ischemia-reperfusion injury, hypertonic lactate, resuscitation

## Abstract

While sudden loss of perfusion is responsible for ischemia, failure to supply the required amount of oxygen to the tissues is defined as hypoxia. Among several pathological conditions that can impair brain perfusion and oxygenation, cardiocirculatory arrest is characterized by a complete loss of perfusion to the brain, determining a whole brain ischemic-anoxic injury. Differently from other threatening situations of reduced cerebral perfusion, i.e., caused by increased intracranial pressure or circulatory shock, resuscitated patients after a cardiac arrest experience a sudden restoration of cerebral blood flow and are exposed to a massive reperfusion injury, which could significantly alter cellular metabolism. Current evidence suggests that cell populations in the central nervous system might use alternative metabolic pathways to glucose and that neurons may rely on a lactate-centered metabolism. Indeed, lactate does not require adenosine triphosphate (ATP) to be oxidated and it could therefore serve as an alternative substrate in condition of depleted energy reserves, i.e., reperfusion injury, even in presence of adequate tissue oxygen delivery. Lactate enriched solutions were studied in recent years in healthy subjects, acute heart failure, and severe traumatic brain injured patients, showing possible benefits that extend beyond the role as alternative energetic substrates. In this manuscript, we addressed some key aspects of the cellular metabolic derangements occurring after cerebral ischemia-reperfusion injury and examined the possible rationale for the administration of lactate enriched solutions in resuscitated patients after cardiac arrest.

## 1. Background

Sudden cardiac arrest (CA) is characterized by the abrupt loss of heart function and organs perfusion and is the third leading cause of death in Europe [1,2,3,4]. Even when circulation is restored and patients are admitted to the hospital, CA remains associated with poor long-term neurological outcome in most survivors [1], constituting a prominent health problem. While acute/chronic coronary disease is the main cause of CA, extended brain damage remains the most important determinant of poor outcome in resuscitated CA patients [5]. With the loss of perfusion pressure, the whole body is exposed to an ischemic injury for a variable amount of time and, eventually, to massive and sudden reperfusion when resuscitation is achieved. Of all organs, the brain is characterized by a very high metabolic demand [6]. In resting conditions, the brain consumes more than 25% of all available circulating glucose and 20% of oxygen, despite it representing only 2% of total body weight [7,8], and the metabolic cost of its development characterizes our species [9]. The brain relies on systemic energetic supply with minimal store capacity and is thus more susceptible to ischemia than other organs [10]. After the restoration of circulation, a complex series of events could further promote the progression of cerebral injuries and could be targeted to reduce cerebral damage. A large number of therapeutic options were successfully tested in the experimental setting; however, none but target temperature management (TTM) was translated into clinical practice [11]. Hence, finding new therapeutic options is a priority in clinical research.

## 2. Mechanisms Involved in Brain Injury after Resuscitation from Cardiac Arrest

A plethora of mechanisms are implicated in the development of anoxic brain injuries after cardiovascular collapse and reperfusion. Immediately after the cessation of circulation, a wide range of events take place, including early depolarization and synaptic dysfunction [12], cerebrovascular autoregulation impairment [13], biochemical modifications in the intracellular and extracellular space [14], and metabolic failure with ATP depletion, metabolic acidosis, free radical formation, and mitochondrial damage [15].

Transmembrane cation gradient is kept in equilibrium in the neurons via energy-dependent extrusion mechanisms, such as the NA^+^/K^+^ ATP-dependent pump. The need to maintain electroneutrality across the membrane is then accompanied by energy failure and an inward directed flow of cations matched by anions, resulting in the concomitant entry of water, ultimately responsible for cellular swelling [16].

Neuronal swelling and cytotoxic edema results from rapid change in osmotic balance between the intra- and extracellular compartments via a strong influx of Na^+^ by multiple depolarizing triggers [17].

With the intracellular increase in both anions and cations, the extracellular milieu becomes depleted in such ions, establishing a new gradient across the blood-brain barrier (BBB) that ultimately leads to an increase in total water volume and brain swelling [18]. While cellular swelling is common between all cellular lines in the brain, astrocytes appear to be more prominently interested than neurons [19]. Necrosis manifests when all possible compensatory cellular mechanisms fail, and the swollen cell ultimately dies [20].

In cardiac arrest, the whole brain is exposed to a variably long period of global ischemia and energy depletion and there is not a defined ischemic core as is the case in stroke. Similar to the penumbra area that surround an ischemic stroke core, the brain injury in CA is not only determined by the severity of the primary injury (i.e., no-flow and low-flow times, corresponding to cardiocirculatory arrest time and cardiopulmonary resuscitation time), but also by the capacity to restore and meet further cellular energetic demand after reperfusion.

When circulation is restored, oxygen supply is not associated with a normalization of all these pathological events, but brain damage could be enhanced by the occurrence of several additional mechanisms; in particular, extended neuroinflammation could be accompanied by an increase in the cerebral blood flow in the first phase after reperfusion, resulting in cerebral hyperemia [21].

The term “reperfusion injury” describes the additional damage that paradoxically occurs after the restoration of blood flow to a previously ischemic tissue. In this phase, cells are exposed to a supplemental damage mediated by the production of reactive oxygen species and inflammatory mediators.

Thereafter, metabolic failure and the increase in intracellular Ca^2+^ influx could impair mitochondrial function and intracellular acidosis becomes persistent. Also, glutamate is released in the extracellular space, producing a state of neuronal hyperexcitability; free radicals are produced in large quantity, apoptotic pathways are upregulated and there is an increase in permeability of the BBB [22], leading to cell death, as summarized in Figure 1.

Interestingly, neuronal subpopulations may show distinct vulnerability to ischemic-reperfusion injuries, with the brainstem neurons being more resistant than cortical and subcortical ones to ischemic stress [23,24,25,26]. However, in a recent series of histologic samples from resuscitated CA patients, similar cellular degeneration hallmarks associated with shrinkage, altered neuronal morphology, and Nissl substance loss were found in most examined brain regions [27].

Strategies to mitigate brain injuries after CA could on one hand be directed to decrease the magnitude of the primary injury by reducing the no-flow time and by CPR optimization (education programs, new technologies), and on the other hand, could attempt to reduce the burden of secondary injuries by modulating the necrotic pathways activated by prolonged exposure to ischemia and subsequent reperfusion.

In summary, the brain’s response to ischemic-hypoxic injury is complex and variable over time and space within the brain, making even more complicated to develop therapeutic strategies that could reduce global cerebral damage, despite many possible targets.

## 3. Lactate as Alternative Energetic Substrate

Since the formulation of the astrocyte to neuron lactate shuttle hypothesis (ANLS) by Pellerin L. and Magistretti PJ. in 1994 [28], lactate shuttling between brain cells was widely discussed. The original theory stated that glutamate release by neurons in the inter synaptic space would be reuptake by astrocytes coupled with extracellular Na^+^ and converted to glutamine via an ATP-dependent reaction. The resulting glutamine could be then released back in the intracellular space and internalized by neurons, where it could be further converted again to glutamate, and thus, regenerate the neurotransmitter pool. The necessity for Na^+^ extrusion from the astrocyte to maintain a favorable gradient for the re-uptake of glutamate requires a positive energetic expenditure, and thus, upregulate the glycolysis leading to lactate accumulation, excretion, and subsequent uptake by the neuron to fuel oxidative ATP production. Moreover, the astrocytes are also the site for glucose storage as glycogen in the brain, where mobilization is ATP-independent and could thus maintain the functionality of the system in situation of limited blood glucose availability.

From a byproduct of cellular metabolism, growing evidence contributed to shift the role of lactate to a key signaling molecule and possible alternative energetic substrate for neurons.

Even though the original theory was challenged on multiple aspects, including the very existence of glutamate-stimulated glycolysis in the astrocytes, the small magnitude of the lactate oxidation and shuttling in in vivo studies, and the role of glycogen derived lactate [29,30,31], the debate remains vivid [32], and the research in neuro-energetics provided a large amount concerning both homeostasis and allostasis of lactate in neuroenergetics [33].

Within the brain, astrocytes and neurons show different metabolic profiles due to the expressions of distinct isoforms of regulating proteins such as lactate dehydrogenase (LDH-1 in the neurons, LDH-5 in the astrocytes), pyruvate dehydrogenase (PDH) and pyruvate kinase (PK) as well as different lactate transporters (MCT2 in neurons and MCT1 and MCT 4 in astrocytes) that may reinforce the idea of neurons as glucose and lactate users compared to more glucose user and lactate suppliers role of the astrocytes [34]. These differences support the idea that a tight metabolic crosstalk is important for proper brain function and that could be impaired in pathological conditions. Once again, despite the incertitude regarding the effective biological correspondence between the ANLS and experimental data, lactate is assuming a crucial role in brain energy homeostasis. Beyond homeostasis, an even greater uncertainty surrounds the role of lactate in pathological situation requiring adaptive mechanisms. Evidence suggests that lactate could maintain cerebral function during hypoglycemia in both healthy humans and insulin-dependent diabetic patients [35,36], and in more recent years lactate enriched solutions were tested in healthy subjects [37], supporting the idea that exogenous lactate could be uptaken and oxidized by the brain during both rest and exercise.

In the context of ischemic brain injury, lactate supplementation impact on brain injury was investigated in a relatively small number of experimental trials. A reduced glutamate and GABA ischemia-evoked release, and EEG improvement was recorded when topically applied on rat cortex after transient four vessels occlusion [38]. In another series of experiments, lactate supplementation protected against neuronal death after oxygen and glucose deprivation in vitro and its intraventricular injection after reperfusion of an occluded middle cerebral artery in vivo reduced infarct size when given immediately after reperfusion and ameliorates neurological outcome (without reducing infarct size) when given one hour after reperfusion [39]. The same group later also confirmed the neuroprotective effect of the intravenous supplementation of lactate in a similar model of middle cerebral artery occlusion [40]. Similarly, intraperitoneal lactate supplementation before or after middle cerebral artery ligation in rats showed strong neuroprotective effects when associated with isoflurane anesthetics, but not phenobarbital [41].

Moreover, lactate enriched solutions exhibit promising effects on other organs, such as the heart, exerting a positive inotropic effect when administered in both healthy humans [42] and patients with acute heart failure [43], and could even mitigate reperfusion injuries after myocardial ischemia [44].

The relative importance of those mechanisms in both physiological and pathological conditions is not yet fully clarified, and the possible role of exogenous lactate in situations of brain ischemic and reperfusion injuries could not be extrapolated from those data. However, in an in vitro experiment of rat hippocampal oxygen deprived cultured slices, lactate wasn’t able alone to support neuronal survival, but whether combined with glucose, exerted a better effect than glucose alone [45], suggesting a profound interplay between different energetic substrate in the injured brain.

In some experimental designs testing different applications of sodium lactate, such as in septic shock, lactate supplementation showed some contrasting results and potentially detrimental effects [46,47], but there is at least one ongoing clinical study that will hopefully clarify its possible application in humans in this context (NCT03528213).

## 4. Lactate Enriched Solutions in Traumatic Brain Injury

Lactate enriched solutions are a relatively large group of fluids that could vary in constituents but are mainly composed by different quantities of lactate and sodium. In the majority of human studies, the investigators have used hypertonic sodium lactate (HSL) solutions with an osmolarity comprised between 0.5 and 1 and most of those studies were conducted in severe brain injured patients (i.e., with a Glasgow Coma Score < 9).

After traumatic brain injury (TBI), animal studies showed that lactate uptake is increased at the injury site [48] and suggest a beneficial effect of the use of HSL in reducing cognitive deficit [49], increasing cerebral blood flow, and decreasing intracranial pressure (ICP) [50].

In a randomized controlled trial including 34 severe TBI patients, Ichai et al. found that 0.5 M HSL infusion compared to an equivalent osmotic load of mannitol was more effective at treating episodes of increased intracranial pressure, with more pronounced and longer lasting reduction in ICP associated with an increase in jugular venous O_2_ saturation, plasmatic glucose and lactate levels, and pH. Treated patients also experienced a better long-term neurological outcome at one year measured with the Glasgow outcome scale [51]. Subsequently, the same group also showed that a continuous infusion of half molar sodium lactate was able to prevent episodes of increased ICP in 60 severe TBI patients. Concomitantly, the group treated with HSL had a smaller cumulative fluid at 48 h, with cumulative sodium balance (delta between intake and output) not different between groups associated with a markedly reduced chloride balance in lactate group [52]. In another study, Bouzat et al. investigated the impact of a 3 h HSL infusion in a series of severe TBI patients (*n* = 15) who received extensive neuromonitoring including ICP, brain oxygen pressure continuous measurement (PbtO_2_), and cerebral microdialysis (CMD). During the HSL infusion, extracellular metabolites were significantly modified, with an increase in cerebral lactate, pyruvate, and glucose, and a concomitant decrease in glutamate, ICP, and PbtO_2_; the authors considered these effects as the shift in cerebral metabolism from a prevalent glucose-driven metabolism to a situation of increased lactate utilization (i.e., “glucose-sparing effects”), while the reduction in brain oxygenation could be secondary to the alkalosis induced by HSL, which increased the affinity of oxygen for hemoglobin [53]. Although this interpretation was largely criticized [54,55], these metabolic changes occurring during the HSL infusion did reinforce the possibility that exogenous lactate supplementation might exert positive effects in situations of severe brain injury. The same group also found an association between baseline metabolic state and the magnitude of the increase in CMD glucose (i.e., higher effects when lactate to pyruvate ratio was >25), suggesting that the modification of brain energetics could be dependent on baseline state. In this study, the authors also found that those modification occurred independently from cerebral blood flow measured with cerebral perfusion scan [56]. More recently, Carteron et al. reported in a mixed population of patients with either TBI or subarachnoid hemorrhage that a 3 h infusion of a 1 M HSL solution was associated with an increase of cerebral glucose and lactate, as well as an increase in middle cerebral artery flow velocities and decreased pulsatility index measured by transcranial Doppler [57], which suggested an increase in cerebral blood flow secondary either to intrinsic vasodilatory properties of lactate or to an augmented cerebral metabolism. Since the CPP/MAP and ICP were stable during the infusion, the forementioned improvement in cerebral perfusion was attributed to lactate supplementation rather than the hypertonicity of the solution. A list of human trials in which sodium lactate infusion was tested is summarized in Table 1.

## 5. Lactate Enriched Solutions beyond Neuro-Energetics

Unfortunately, cerebral metabolic rate and cerebral metabolism are difficult to measure both in experimental and clinical settings due to the peculiarity in blood supply and drainage of the brain; the situation is further complicated considering regional and cellular specific differences in metabolism, tolerance to ischemia/reperfusion, and regenerative capacities. As such, no conclusive statement could be made upon the cellular metabolic effects induced by hypertonic lactate supplementation in the heterogeneous population of brain injured patients. Nevertheless, many intrinsic properties of HSL beyond its impact on metabolism suggest it as a possible therapeutic option following cardiac arrest to ameliorate perfusion and mitigate brain damage, as summarized in Figure 2.

Firstly, lactate enriched solutions have a high osmolarity and induce an increase in cerebral blood flow both during resuscitation and after ROSC [58]. Moreover, in a study by Sekhon et al., heterogeneous levels of ICP were recorded in a population of patients who underwent invasive neuromonitoring including continuous recording of ICP, showing that episodes of increased ICP might occur in CA survivors in the hours following resuscitation. In this population, 11% of the recorded ICP values were >20 mmHg, accounting for an exposure to intracranial hypertension equal to 22% of the total measured time, and two patients among the 10 included had a refractory ICP increase despite maximal medical treatment [59]. For these patients in whom episodes of increased ICP or sustained intracranial hypertension occurs, HSL exhibits a more favorable profile to mitigate these shifts in intracranial pressure compared to mannitol, which could cause dehydration due to osmotic diuresis and arterial hypotension [60], and is associated with a rebound increase in ICP after discontinuation [61].

Secondarily, lactate was found to reduce the impact of neuronal glutamate-related excitotoxicity in an experimental setting. When L-lactate was added to glutamate perfused continuously into the cortex of unanesthetized rats, a reduction in lesion size without concomitant reduction in extracellular glucose. Interestingly, the effect was not present when only D-lactate was added to the medium [62]; increased glutamate after the anoxic injury might increase the number of neuronal cells evolving towards apoptosis and the risk of seizures [63]. In addition, the tonicity in lactate enriched solutions is also provided by a high sodium load, which might be helpful to mitigate the massive shift in Na^+^, partially responsible for cerebral cytotoxic oedema. To provide the same amount of sodium with differently concentrated hypertonic saline, an equivalent load of Cl^−^ would be necessary, which would result in hyperchloremic metabolic acidosis and possibly increased incidence in renal failure. Furthermore, intracellular hyperchloremia exert a direct toxicity on the neurons, causing hyperexcitability and altering neuronal action potential threshold [64].

On the opposite, exogenous administration of lactate, normally as a racemic mixture of d- and l- lactate, constitutes a weak base and could induce metabolic alkalosis if given in large quantities or during prolonged periods. However, in the specific setting of resuscitated cardiac arrest, patients usually present with extremely deep metabolic and respiratory acidosis, which correction might take hours and is dependent on the cardiac pump ability to recover and to restore tissue perfusion as well as on proper ventilation to eliminate the excess of CO_2_. In theory, buffering extremely deep acidosis could improve response to catecholamines, increase venous return, and re-establish favorable pharmacokinetics for most administered drugs. The debate on whether a buffer solution might be beneficial or harmful in cardiac arrest was ongoing for decades, leading firstly to inclusion of sodium bicarbonate in resuscitation guidelines, and then in discouraging it [65,66]. In contrast to sodium bicarbonate, sodium lactate dissociation does not lead to an increase in blood and tissular CO_2,_ and thus, does not contribute to worsen intracellular acidosis.

Importantly, hypernatremia, hypokalemia, and metabolic alkalosis are among the anticipated side effects of this treatment and might exert a particularly detrimental effect in the context of resuscitated CA patients so that a careful administration under strict monitoring seems to be a proper strategy and must be always considered.

To sum up, beyond its possible energetic beneficial properties, HSL presents a good asset of properties suited for treatment in the context of cardiac arrest and should be evaluated as possible therapeutic strategy in this context.

## 6. Conclusions and Future Perspectives

Immediately following cardiocirculatory arrest and for several hours after reperfusion, a complex series of event participate in maintaining and further increase brain damage and cellular dysfunction. Lactate enriched solutions can enhance cerebral metabolism, mitigate metabolic acidosis, limit cerebral oedema, and ameliorate cerebral perfusion, and thus, seem to be a promising therapeutic option in this context. Clinical studies are warranted to enlighten the role of lactate enriched solutions in CA patients.

## Figures and Tables

**Figure 1 cells-10-01714-f001:**
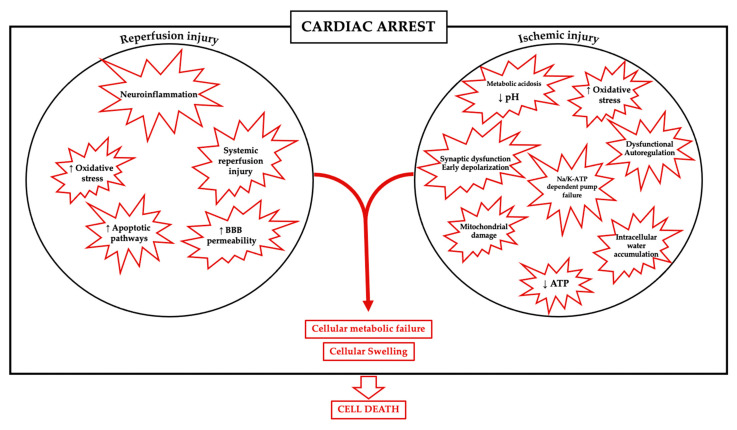
Mechanisms involved in brain injury after ischemia-reperfusion.

**Figure 2 cells-10-01714-f002:**
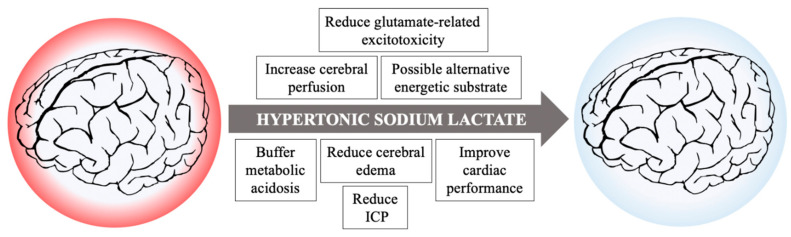
Possible beneficial actions of hypertonic sodium lactate infusion in resuscitated patients after cardiac arrest.

**Table 1 cells-10-01714-t001:** Studies in humans with sodium lactate supplementation.

Author	Study Type	Condition	*n*	Primary Outcome	Additional Therapies Given/Measurement	Dose	Total Length of Administration	M	Osm/L	Reported Severe Adverse Events
Van Hall 2009 [30]	Experiment study in vivo	Healthy subjects	6	Establish if systemic lactate is an energy source for human brain in vivo and comparison with skeletal muscle	Lactate infusion tested in both resting state and during exercise	Prime: 0.8 µmol/Kg/min for two hours then 2.4 for 90 min and 8 for 30 min Infusion: 50 µmol/Kg/min for 20 min and then 36 µmol/Kg/min for 100 min	90 min (rest) and 30 min (exercise)	-	-	-
Ichai 2009 [38]	RCT	Severe TBI	34	Efficacy in reducing ICP during intracranial hypertension (ICP > 25 mmHg for more than 5 min without noxious stimulation)	Hyperosmolar, isovolumic infusion of Mannitol 20%. Treatment with both drugs was allowed based on the efficacy of the first therapy	1.5 mmol/Kg over 15 min	Bolus, max 2 × per episode, max 3 episodes/patients	0.5	1100	none
Ichai 2013 [39]	RCT	Severe TBI	60	Effect of continuous lactate infusion (Totilac^®^, 48 h) to prevent raised ICP episodes	0.9% sodium chloride solution isovolumic infusion in the control group	4.16 µmol/Kg/min	48 h	0.5	1100	none
Nalos 2014 [32]	RCT	AHF	40	Cardiac output at 24 h measured by TTE	Hartmann’s solution for control group	1.5 mmol/Kg over 15 min then 8.3 µmol/Kg/min	24 h and 15 min	0.5	1000	none, but significant changes in electrolytes levels)
Bouzat 2014 [40]	Prospective Phase II clinical trial	Severe TBI	15	Effect of lactate infusion on CMD concentration of lactate, pyruvate, and glucose	-	40 µmol/Kg over 60 min, then 30 µmol/Kg for 120 min**	3 h	1	2000	none
Quintard 2015 [43]	Prospective interventional trial	Severe TBI	24 *	Effect of lactate on cerebral energetics according to baseline CMD LPR and CBF quantified by perfusion CT	-	30–40 ** µmol/Kg/min	3 h	1	2000	none
Nalos 2018 [31]	Randomized crossover study	Healthy subjects	10	Determine metabolic and cardiac effect of lactate infusion by TTE and blood gas analysis	3% sodium chloride isovolumic infusion	1.5 mmol/Kg over 20 min, then 16.6 µmol/Kg/min	80 min	0.5	1000	-
Carteron 2018 [44]	Prospective interventional trial	Severe TBI/SAH	13/10	Analyze changes in cerebral perfusion (TCD) and metabolism (CMD) during lactate infusion	-	30 µmol/Kg/min	3 h	1	2000	none

M: Molarity; RCT: Randomized controlled trial; TBI: traumatic brain injury; AHF: acute heart failure; TTE: trans thoracic echocardiography; CMD: cerebral microdyalisis; LPR: lactate/pyruvate ratio; CBF: cerebral blood flow; SAH: subarachnoid hemorrhage; TCD: transcranial Doppler. * 15 patients included also in the study by Bouzat et al. in 2014. ** Dose adjusted to reach a steady arterial lactate concentration of 4–5 mmol/L.

## Data Availability

Not applicable.

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
