# Peer review of "Brain Protection after Anoxic Brain Injury: Is Lactate Supplementation Helpful?"

_cells, 2021, doi:10.3390/cells10071714_

Round 1

Reviewer 1 Report

In this article, the authors has had an overview on the potential benefits of lactate supplementation in brain damage post anoxic injuries/events such as cardiac arrest. The paper is overall succinct and well written. I have few minor comments that could improve the quality of paper for publications in Cells:

Comment 1. For section 2, please provide a figure or diagram that shows various potential mechanisms involved in brain injury after cardiac arrest or post resuscitation. This can be more informative in addition to the section that you provided.

Comment 2. In Table 1, the authors have included clinical studies in which lactate supplement was given in either healthy subjects or patients with mostly TBI; however,  the outcome measures/results of these clinical studies are lacking in this table. Additionally, it should be mentioned whether these individuals were being treated with other study drugs or relevant treatment at the same time. The table needs a significant revision to add these outcomes/issues. This can help readers understand whether lactate supplements were safe or effective in these subjects.

Author Response

In this article, the authors has had an overview on the potential benefits of lactate supplementation in brain damage post anoxic injuries/events such as cardiac arrest. The paper is overall succinct and well written. I have few minor comments that could improve the quality of paper for publications in Cells:

We would like to thank the Reviewer for her/his comment.

Comment 1. For section 2, please provide a figure or diagram that shows various potential mechanisms involved in brain injury after cardiac arrest or post resuscitation. This can be more informative in addition to the section that you provided.

A schematic representation of the section 2 has been added to the manuscript.

Comment 2. In Table 1, the authors have included clinical studies in which lactate supplement was given in either healthy subjects or patients with mostly TBI; however, the outcome measures/results of these clinical studies are lacking in this table. Additionally, it should be mentioned whether these individuals were being treated with other study drugs or relevant treatment at the same time. The table needs a significant revision to add these outcomes/issues. This can help readers understand whether lactate supplements were safe or effective in these subjects

We agree that the table should have been extended to provide more informations. We have reviewed the table and expanded by adding four columns, including the study type, the primary outcome, other treatments/measurements and reported severe adverse events.

Reviewer 2 Report

This is a well written manuscript of interest to the readership of Cells. I have some comments that would help the reader.

  1. It would be helpful to have a figure to summarize the mechanisms discussed in Section 2.
  2. Sentence in lines 106-111 is missing a verb or clause and is too long.  I would suggest breaking up the sentence.
  3. Discussion and a figure discussing how lactate supplementation acts at the molecular level would aid the reader.
  4. The following sections are missing:

    Author Contributions:

    Funding:

    Institutional Review Board Statement:

    Informed Consent Statement: 

    Data Availability Statement:

    Conflicts of Interest: 

Author Response

We would like to thank the Reviewer for her/his comments

  1. It would be helpful to have a figure to summarize the mechanisms discussed in Section 2.

A schematic representation of section 2 has been added to the manuscript

  1. Sentence in lines 106-111 is missing a verb or clause and is too long.  I would suggest breaking up the sentence.

Thank you for your comment: the sentence was indeed missing a part and has been corrected and divided.

  1. Discussion and a figure discussing how lactate supplementation acts at the molecular level would aid the reader.

Thank you for your suggestion: we agree that the information regarding the effect of lactate at a molecular level could be more exhaustive and complex. Nevertheless, our intent was to present a cellular based rationale for the possible application of hypertonic lactate solutions in the clinical setting more than a comprehensive review of its mechanisms, but we have provided consistent references for the reader to explore the debate around the role of lactate at the cellular level.

  1. The following sections are missing:

Author Contributions:

Funding:

Institutional Review Board Statement:

Informed Consent Statement: 

Data Availability Statement:

Conflicts of Interest: 

Sections have been added to the text.

Round 2

Reviewer 1 Report

The authors has replied well to all my comments.